# Reasoning and Tools for Human-Level Forecasting

**Elvis Hsieh**[*]     **Preston Fu**[*]     **Jonathan Chen**[*]

UC Berkeley
{htelvis92,prestonfu,jonchen25}@berkeley.edu

## Abstract

Language models (LMs) trained on web-scale datasets are largely successful due to their ability to memorize large amounts of training data, even if only present in a few examples. These capabilities are often desirable in evaluation on tasks such as question answering but raise questions about whether these models can exhibit genuine reasoning or succeed only at mimicking patterns from the training data. This distinction is particularly salient in forecasting tasks, where the answer is not present in the training data, and the model must reason to make logical deductions. We present Reasoning and Tools for Forecasting (RTF), a framework of reasoning-and-acting (ReAct) agents that can dynamically retrieve updated information and run numerical simulation with equipped tools. We evaluate our model with questions from competitive forecasting platforms and demonstrate that our method is competitive with and can outperform human predictions. This suggests that LMs, with the right tools, can indeed think and adapt like humans, offering valuable insights for real-world decision-making.

## 1   Introduction

Forecasting is an essential tool today, playing a critical role in government, corporate, and personal decision-making. Weather forecasting provides essential information for agriculture, natural disaster preparedness for governments, and travel plans for individuals. During the COVID-19 pandemic, lockdown policies were largely determined by forecasts, which were required to be sufficiently accurate due to their global impact [9]. Forecasting methodologies fall into two main categories [23]: statistical and judgmental. Statistical forecasting leverages time-series modeling and excels with abundant data under stable conditions. Conversely, judgmental forecasting, which we refer to simply as "forecasting," typically relies on human expertise, integrating historical data, domain knowledge, and intuition to make predictions, and is particularly useful when data are sparse or conditions are volatile. By nature, forecasting requires not only accuracy but also the ability to continuously adapt to dynamic data streams. This is where traditional LMs often struggle: timely data updates may cause predictions to change considerably and past data to be irrelevant.

## 2   Related Work

**Information retrieval**   Reliable and accurate predictions are largely dependent on the information available to the predictor. This is especially the case of LMs, which are trained on data preceding a knowledge cutoff and have been shown to perform better with information retrieval [20].

Language models model the likelihood $p_\theta(y_i|x, y_{<i})$ for input sequences $x$ and target sequences $y$. Retrieval-augmented generation (RAG) [12] proposes augmenting this approach with non-parametric memory, i.e. retrieving the top-$k$ text documents $z$ via $p_\eta(z|x)$ and conditioning the generator on

---

[*]Equal contribution

38th Conference on Neural Information Processing Systems (NeurIPS 2024).

Figure 1: RTF: High-level agent oversees low-level agents, each equipped with distinct toolkits and data/document to accomplish various tasks, including API calling and Python simulation.

Table 1: Performance of different models with the same prompt on forecasting questions. "Base LM" refers to {GPT-4o, 4, 3.5, Llama 3}. "Acc" is accuracy, and "Std" is ensemble standard deviation.

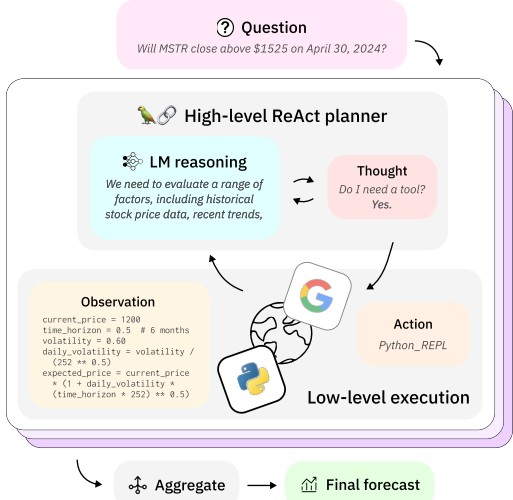

| Method | Brier ↓ | Acc % ↑ | Std ↓ |
|---|---|---|---|
| Crowd | 0.172 | 73.8 | |
| RTF Median of 3 | **0.169** | 72.4 | 0.092 |
| RTF Mean of 3 | 0.170 | **73.9** | 0.092 |
| RTF Sampled | 0.180 | 71.6 | |
| Halawi et al. [10] GPT-4o | 0.177 | 68.7 | |
| GPT-4o | 0.210 | 65.5 | |
| Base LM Mean | 0.218 | 62.9 | 0.150 |
| Base LM Median | 0.228 | 61.3 | 0.150 |
| Llama 3 | 0.256 | 56.2 | |
| GPT-3.5 | 0.261 | 53.5 | |
| GPT-4 | 0.265 | 54.8 | |

the retrieved passages, $p_\theta(y_i|x, z, y_{<i})$. In a forecasting context, RAG enables us to search for relevant documents $z$ that may contain timely information about the forecasting task $x$ not present in the training data. Therefore, we propose to equipping our forecasting agent with tools to retrieve up-to-date information beyond its knowledge cutoff.

**Prior approaches to LLM forecasting** LMs have shown strong, interpretable reasoning capabilities [3, 2, 1], and have seen adoption across a wide variety of domains, even in cognitively demanding tasks. LMs have shown success across a large number of benchmarks, but previous work has shown that LMs predictly emit memorized training data [6, 8]. As a result, it seems plausible that success on these benchmarks merely corresponds to memorization. On the other hand, forecasting has proven to be a difficult task for LLMs. For example, Zou et al. [27] proposed a benchmark to use neural networks to automate prediction in prediction markets; the authors found that while language models can be trained to improve their performance on forecasting tasks, their accuracy remains significantly below those of human experts.

Current methods aim to improve the accuracy of LLM forecasting by fine-tuning and scratchpad prompting [16, 10, 24] or ensembling [5, 18] in an attempt to approach human-level forecasting. However, fine-tuning requires extensive computational resources and dataset preparation, resulting in a costly process that limits scalability, especially for specialized applications such as forecasting. Concurrent work [17] benchmarks LLMs' forecasting capabilities using the GleanGen prediction market, an internal tool at Google. However, this approach did not accurately reflect real human crowd prediction distributions, and it relied on PaLM2 [4], which performs worse than GPT models. We propose a zero-shot tool-usage LLM framework without costly fine-tuning and tedious scratchpad-format prompting.

**Ensembles** Leveraging multiple LLM agents has demonstrated strong performance on a variety of tasks, and improve performance beyond that of a single agent [22, 13]. Schoenegger et al. [18] has shown that taking ensemble sizes up to 36 outperforms any individual forecasting agent. Recent work in tool learning has implemented task planning and execution with separate agents [21, 19]. We propose bridging this gap with a hierarchical structure to facilitate cooperation between high-level reasoning and low-level execution agents, and demonstrate that a small ensemble suffices for human-level performance without costly fine-tuning and tedious scratchpad format prompting.

# 3 Reasoning and Tools for Forecasting

Forecasting is a complex task solving environment, for which we would like to where we leverage a frozen LM $p_\theta$ as reasoning. Successful forecasting agents rely on the most up-to-date information, and accordingly operate as agents that collect observations $\mathbf{o}_t \in \mathcal{O}$ and take actions $\mathbf{a}_t \in \mathcal{A}$. The observation space $\mathcal{O}$ is natural language, as collected from the prompt itself or information on the internet. The agent's actions are distributed according to $\mathbf{a}_t \sim \pi(\mathbf{a}_t|\mathbf{c}_t)$, where $\mathbf{c}_t = (\mathbf{o}_1, \mathbf{a}_1, \ldots, \mathbf{o}_{t-1}, \mathbf{a}_{t-1})$ is the context to the agents.

Our proposed approach $\pi$ satisfies the following criteria:

- It is **simple, scalable, and time-invariant**. As we consider different datasets of forecasting questions or language models at least as capable as the current state-of-the-art, we would like our approach to work at least as well.

- It can produce comprehensive responses through zero-shot prompting from factual information, which can be used to **reliably support decision-making** in downstream scenarios.

- These responses should be **consistent**, i.e. they should correctly synthesize the up-to-date information the model collects.

It's shown that CoT prompting, even with in-context examples, can iteratively hallucinate to produce incorrect responses on complex tasks [25]. CoT satisfies (i) but neither (ii) nor (iii). We find that CoT's lack of interaction with the environment (i.e. sole reliance on its training data) limits its reasoning abilities and over-emphasizes irrelevant information. Yao et al. [25] proposes ReAct for this setting: $\mathcal{A} = \{\text{search}, \text{lookup}, \text{finish}\}$, and observations $\mathbf{o}_t$ from search and lookup are collected from $\mathcal{O} \subseteq$ Wikipedia web API. The context is then augmented a thought $\hat{\mathbf{a}}_t \sim p_\theta(\hat{\mathbf{a}}_t|\mathbf{c}_t)$ that composes information about the existing context. This method has shown to significantly enhance the model's ability to refine its responses continuously, reducing the likelihood of erroneous outputs due to lacking critical context information. Vanilla ReAct satisfies (i); as part of our framework, we show that it can additionally satisfy (ii) and (iii).

**Hierarchical planning** We define $\pi$ by an aggregate of a collection of hierarchical ReAct agents with tools for real-time data retrieval and simulation, expanding $\pi$'s observations $\mathbf{o}_t$ collected from $\mathcal{O} \subseteq$ Google Search API and Python interpreter.

We propose hierarchical ReAct planning, where a LM agent acts as a high-level planner for handling abstract logic and forecasting principles based on the outputs collected from the low-level agents (Figure 1). When LLMs handle API directly with individual agents, it can consume a large portion of the context window. We delegate the reasoning and API calling to specialized agents to enhances efficiency, conserves tokens, and allows for more complex operations. The high-level agent interacts with the low-level agent by invoking it as a tool. We wrap API tools with another ReAct agent to form the low-level agent, which significantly increases API call success rates due to its self-correction mechanism [25]. Both classes of agents are implemented with GPT-4o backbones.

# 4 Experiments

## 4.1 Setup

**Models and data** Jin et al. [11], Zou et al. [26] have proposed forecasting benchmarks to assess models' forecasting abilities, simulating forecasting by leveraging that models are only trained up to a cutoff date. However, these benchmarks, consisting of questions that resolved in 2022, are now outdated for evaluating the performance of models such as GPT-4o due to answer leakage in training data (knowledge cutoff October 2023; see Appendix A.1).

We curated the dataset on April 15, 2024, when we scraped the platform for questions resolving within the next two weeks and corresponding human crowd predictions. We then filtered out vague questions, and ran every prediction method on these questions, enabling a fair comparison between each method and the human crowd. To prevent answer leakage from the Google API, we set the search range to prior to this date. Our final dataset consisted of 201 questions spanning across 9 diverse categories (see Apendiex B).

None of our baselines have direct access to prediction market data, and empirically we found that this information was never scraped via Google search. That is, the prediction given by the ensemble of agents relies on only the agents themselves, with no human crowd influence. (By contrast, if deployed in the real world, this approach could benefit from incorporating the current human crowd performance as an input to the prediction due to the wisdom-of-crowds effect. Indeed, we observe in our experiments that human crowds are fairly well-calibrated.)

**Performance metrics**  Our $n$ forecasting questions have true outcomes $o_i \in \{0, 1\}$ and probabilistic forecasts $f_i \in [0, 1]$. We evaluate our forecasts using Brier scores [7], i.e. $\frac{1}{n} \sum_{i=1}^{n} (f_i - o_i)^2$, and accuracy, i.e. $\frac{1}{n} \sum_{i=1}^{n} \mathbf{1}\{\mathbf{1}\{f_i > 0.5\} = o_i\}$.[2] [3] In case LMs decline to give numerical answers, the question is dropped over all methods when evaluating scores.

**Baselines**  In Table 1, we compare RTF ensemble to multiple baselines: (a) crowd scores given by the current traded values on Manifold Markets (see Appendix A.2), (b) scratchpad prompting, ensemble, and fine-tuning [10], and (c) base models from different providers.

## 4.2  Results and Observations

Table 1 demonstrates that RTF significantly improves over CoT and scratchpad with fine-tuning. We also achieve comparable Brier score (0.169 vs. 0.172) and superior accuracy (73.9% vs. 73.8%) compared to human predictors using the median and mean of our ensemble, respectively.

We also demonstrate that ensembles for RTF yield better performance than individual agents (Brier 0.169 vs. 0.180). However, this is not the case for base LMs (Brier 0.218 vs. 0.210 for GPT-4o). Base LMs tend to produce higher-variance outputs (standard deviation in ensemble size 4 of 0.150) compared to our better-calibrated ReAct agents (standard deviation in ensemble size 3 of 0.092), which satisfied (iii) as defined in Section 3.

Ensembles only contribute to the final performance if each ensemble member is already sufficiently calibrated. Indeed, Brier scores given by randomly sampling our ReAct ensemble outputs, "React Sampled" in the table, achieved a score of 0.180, far better than was achieved by any of the base methods (which, aside from GPT-4o, perform worse than guessing 0.5 every time by Brier score).

**Ablation study**  To demonstrate the effectiveness of our introduced components, we conduct the ablation study. We showed each component is necessary for the fully functioning RTF framework.

- **ReAct:** RTF itself without adequate guidance from ReAct struggles to properly use the tools provided by our low-level agents, which leads to misguided lines of reasoning that cascade downstream. This is consistent with the observation (B) in [25], where groundedness and trustworthiness come at the cost of higher reasoning error rates.
- **Hierarchical Planning:** Empirically, without the cooperation of high- and low-level agents, a single agent fails to call APIs and perform necessary reasoning, as it exhausted available tokens on API schemas. In our experiments, the single-agent approach frequently encountered time-out errors or exceeded rate limits when handling complex queries.

**Qualitative analysis**  While the baselines systematically evaluate multiple considerations, they do not consider interactions between these considerations. Empirically, we find in our samples that the prompting style we present is useful in generating a wide variety of arguments and providing reasonable estimates for how to weight each of those arguments. On the other hand, we see that this same prompt GPT-4o directly does this calibration in a sequential manner to update its final estimate, which may result in over- or under-estimate based on the recency of its considerations. In general, we find that RTF yield human-like reasoning trajectories, showing the robustness of interactive decision making, supporting goal (ii) from Section 3 (see Appendix D).

**Calibration index**  In Table 2, we evaluate our methods by calibration index, which compares binned forecast probabilities to observed outcomes. A well-calibrated model means that if a forecast

---

[2]The optimal strategy to minimize Brier scores is to forecast $f_i = \mathbb{P}(o_i = 1)$, so this scoring metric is unbiased. It is typical to compare Brier scores to 0.25, which can be achieved by $f_i = 0.5$ for all $i$.

[3]Accuracy denotes whether $f_i$ and $o_i$ are on the same side of 0.5.

Table 2: Calibration index for different methods.

| Method | Crowd | ReAct Mean | ReAct Median | ReAct | GPT-4o | GPT-4 | Llama 3 |
|---|---|---|---|---|---|---|---|
| **Calibration Index** | 0.0101 | **0.0129** | 0.0137 | 0.0164 | 0.0194 | 0.0290 | 0.0301 |

predicts an event with a certain probability, the event should occur approximately that fraction of the time over many predictions.

We calculate the calibration index as

$$CI = \frac{1}{N} \sum_{k=1}^{K} N_k (f_k - o_k)^2,$$

where $N$ is the total number of forecasts, $N_k$ is the number of forecasts in bin $k$, $f_k$ is the mean forecast probability in bin $k$, and $o_k$ is the observed probability with which events occur in bin $k$. We select bins as the $K$-quantiles of the forecasts.

Comparing GPT-4o and React Mean, we see a significant decrease in calibration index (0.0194 vs. 0.0129), which shows that ensembling with ReAct not only increases forecasting accuracy, but also more accurately measures the specific magnitudes with which events occur.

## 5 Conclusion

We present Reasoning and Tools for Forecasting, a framework to leverage LMs' reasoning capabilities by interacting with the latest information. It is competitive with the predictive capabilities of human forecasters on forecasting platforms. The RTF synthesizes information through a structured decision-making process, ensuring that the predictions are both current and relevant. Additionally, while previous work has shown that ensembling can improve prediction accuracy, a carefully calibrated smaller set of models is often more cost-effective than larger ensembles. By advancing LMs' abilities to reason and dynamically interact with new data, RTF offers a robust tool for real-world decision-making for tasks like forecasting.

**Limitations** The evaluation dataset is based on prediction market data and popular questions rather than domain-specific questions. This facilitates a comparison with crowd prediction performance, but may not fully capture the nuances of more specialized domains. In addition, the Google Search API retrieves a list of URL links, titles, and text snippets, but lacks nuanced, context-specific understanding. As we can observe qualitatively (see Appendix D.2), interpreting incoherent observations from the API can be challenging.

**Acknowledgments** We appreciate the inspiration from Prof. Jacob Steinhardt's amazing forecasting class at UC Berkeley. We thank OpenAI for granting API credits.

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

# A  Models and Knowledge Accuracy

## A.1  Models

Table 3: Models we evaluated

| Model | Knowledge Cutoff | Evaluation Cost |
|---|---|---|
| GPT-4o | Oct 2023 | $0.005/1K tokens |
| GPT-4-Turbo | Apr 2023 | $0.01/1K tokens |
| GPT-3.5-Turbo | Sep 2021 | $0.0005/1K tokens |
| Llama-3-7B | Mar 2023 | One GPU |

We list the details of models we evaluated in Table 3, where the cutoffs have been retrieved from the model cards. For GPT models, we run them via the OpenAI API. We host Llama-3-7B on a single NVIDIA TITAN RTX 24GB via Ollama for roughly 0.5 GPU-hours. All other approaches are run through the OpenAI API, for roughly 1 hour per naive baseline, 6 hours for our reproduction of [10], and 2.5 hours for our proposed method. For GPT models, we use temperature 0.5 for all the experiments.

[10] finds that GPT-3.5 and GPT-4 do not have leakage due to post-training. We find that the same is true of GPT-4o and Llama-3-7B: prompting with "Answer this question without searching the web: Who was appointed to the Governor-General of Australia in 2024?" yielded a statement about its cutoff date, whereas the correct answer was given when prompted for the answer in 2019.

## A.2  Crowd Predictions

On Manifold Markets, players make bets on the outcome of various events where the prices of bets are determined by a current aggregate of crowd predictions, which are prices in $[0, 1]$. As bets are made, the prices are adjusted by their automated market-makers [14]. As shown in [15], the crowd prediction is a strong baseline and consistently outperform top forecaster in the prediction market.

# B  Dataset

## B.1  Questions

Our final dataset consisted of 201 questions from Manifold Markets. These question were all resolved after April 15, 2024, which was the knowledge cutoff date for our low-level agent supporting the Google Search API. We include a subset of the dataset for reference.

From Manifold Markets, we initially filtered for questions that resolve between April 16, 2024 and May 15, 2024, inclusive. Then, to improve the quality of our questions, we filtered the question through prompting GPT-4 to return "Yes" if (i) it is possible to respond to the question with a yes or no answer and (ii) the question asks about an external event from the perspective of the majority of askers and respondents, and "No" otherwise. Finally, after the markets have resolved, we re-collect data using the API to extract the answers and compute Brier scores and accuracies. In the future, researchers can use our questions data for forecasting using LMs and information retrieval tools with cutoff dates before April 15, 2024 (see Section 4.1).

## B.2  Knowledge Evaluation by Category

We show the diversity of our dataset in Table 4, with categories determined by GPT-3.5. Due to the popularity of Economics & Business and Politics & Governance questions on the forecasting platform, we have higher proportions of data in those categories.

Table 4: Category frequencies

| Count | Category |
|---|---|
| Economics & Business | 68 |
| Politics & Governance | 34 |
| Science & Tech | 29 |
| Arts & Recreation | 29 |
| Sports | 16 |
| Security & Defense | 13 |
| Healthcare & Biology | 5 |
| Environment & Energy | 4 |
| Social Sciences | 3 |
| Total | 201 |

## C  Forecasting Principles Guidance

Our system prompt requested that the output satisfy elementary forecasting principles. Due to its length, the full prompt will be released along with the codebase and dataset. The forecasting principles are as follows:

- You may find relevant time series data online. If the data is recent, you can use zeroth-order or first-order approximations (but for longer time horizons, recall that these may break down due to saturation effects).

- Don't ignore base rates for low-probability events that haven't happened, but rather attempt to estimate those probabilities to inform your estimate.

- Attempt to approximate the same quantity in multiple different ways. You are welcome to split your reasoning into multiple sections, where in each section you describe how you reached an estimate independently. If your estimates in these sections are very far off, reason through why the estimates are so different, potentially update those estimates, and theorize how confident you are in each estimate. Then, combine those forecasts using a weighted or trimmed mean or median.

- Some data can be modeled well using a common distribution, e.g., normal, log-normal, power law, or Poisson distribution (but this is by no means a guarantee). If there's a good reason to believe this is the case, you should use information about those distributions.

## D  Prompts and Workflows

Due to length, we have only included one full sample below. In general, we observe that RTF yields more robust CoT-style outputs, whereas base LMs use more linear reasoning that may result in outputs unnecessarily biased towards specific considerations.

### D.1  Prompt

```
Will ETH close above 3700?
Background: None
Resolution criteria: None
Closure time: April 30, 2024
```

For reference, this prompt resolved as NO (0).

### D.2  RTF Output

```
[Entering new AgentExecutor chain...]
To forecast the probability that ETH (Ethereum) will close above $3700 on April 30, 2024, I'll follow a
     structured approach:

1. **Reference Classes and Base Rates**:
    - Historical price trends of Ethereum.
    - Historical performance of similar cryptocurrencies.
    - General market conditions and trends in the cryptocurrency market.
```

```
2. **Recent News and Distinctive Features**:
   - Recent developments in Ethereum technology.
   - Regulatory news affecting cryptocurrencies.
   - Broader economic conditions impacting the cryptocurrency market.

3. **Statistical Modeling**:
   - Use historical price data to model potential future price movements.
   - Consider volatility and distribution patterns typical for Ethereum.

### Step-by-Step Analysis

#### 1. Reference Classes and Base Rates
First, let's look at the historical price data of Ethereum. We want to understand how often Ethereum has
    closed above $3700 in the past and under what conditions it has done so.

Thought: Do I need to use a tool? Yes
Action: Google Search Snippets
Action Input: "historical price data of Ethereum"
To forecast the probability that ETH (Ethereum) will close above $3700 on April 30, 2024, I'll follow a
    structured approach:

1. **Reference Classes and Base Rates**:
   - Historical price trends of Ethereum.
   - Historical performance of similar cryptocurrencies.
   - General market conditions and trends in the cryptocurrency market.

2. **Recent News and Distinctive Features**:
   - Recent developments in Ethereum technology.
   - Regulatory news affecting cryptocurrencies.
   - Broader economic conditions impacting the cryptocurrency market.

3. **Statistical Modeling**:
   - Use historical price data to model potential future price movements.
   - Consider volatility and distribution patterns typical for Ethereum.

### Step-by-Step Analysis

#### 1. Reference Classes and Base Rates
First, let's look at the historical price data of Ethereum. We want to understand how often Ethereum has
    closed above $3700 in the past and under what conditions it has done so.

Thought: Do I need to use a tool? Yes
Action: Google Search Snippets
Action Input: "historical price data of Ethereum"
Ethereum Price History. Daily. Weekly. Monthly. 3/28 ... Historical Data. Frequently Asked Questions (FAQs)
    ... What Does Ethereum's Market Cap History Reveal About... Jan 8, 2023 ... Price history of ETH: On
    August 7, 2015, the first Ethereum transaction was carried out using the transaction hash of block
    46147. Since its... Ethereum's (ETH) highest price in late 2021 was much higher than any other price
    seen in a chart that looks at the previous 5 years as well as 2023. Track the latest Ethereum price,
    market cap, trading volume, news and more with CoinGecko's live ETH price chart and popular...
    Features real-time (live) charts, ethereum blockchain, news and videos. Learn about ETH, the current
    hashrate, crypto trading and more. Dec 18, 2016 ... There is no canonical ETH or BTC price, each
    exchange has its own. Exchange may or may not collect historical data, and granularity of such data
    ... Sep 1, 2017 ... Ethereum price history API? I am working on a personal project to develop graphs
    of the 7 day, 1 month, 3 month, etc. price history of ethereum. Jan 30, 2022 ... ... ethereum-get ...
    ETH price as an integer on an IoT device like the ESP8266  3  Read all historical price data of a
    Chainlink price feed in... Mar 21, 2023 ... Historical tick and granular OHLC price data for Ethereum
    tokens based on DEX swaps. Low resolution (can only get hourly or daily price data... Ethereum Price
    History. Crypto prices are volatile in the sense that they can explode to the upside or blow up your
    life savings. Since its launch, ETH has made...]The search results indicate that there are multiple
    sources for historical Ethereum price data. For a thorough analysis, we'll need to pull specific
    historical data points. Let's proceed by gathering detailed historical price data for Ethereum.

#### 1. Reference Classes and Base Rates (Continued)
To get a sense of the base rate, we need to know how often Ethereum has closed above $3700 in the past and
    under what conditions.

Thought: Do I need to use a tool? Yes
Action: Google Search Snippets
Action Input: "Ethereum historical price data 2021 2022 2023"
The search results indicate that there are multiple sources for historical Ethereum price data. For a
    thorough analysis, we'll need to pull specific historical data points. Let's proceed by gathering
    detailed historical price data for Ethereum.

#### 1. Reference Classes and Base Rates (Continued)
To get a sense of the base rate, we need to know how often Ethereum has closed above $3700 in the past and
    under what conditions.

Thought: Do I need to use a tool? Yes
Action: Google Search Snippets
Action Input: "Ethereum historical price data 2021 2022 2023"
Ethereum's (ETH) highest price in late 2021 was much higher than any other price seen in a chart that looks
    at the previous 5 years as well as 2023. Jan 8, 2023 ... Ethereum was first conceptualized in 2013.
    Ether (ETH) is the native cryptocurrency of Ethereum. On November 9, 2021, Ethereum briefly attained
    ... Frequently Asked Questions (FAQs). How Has Ethereum's Price History Influenced the Cryptocurrency
    Market? Nov 18, 2022 ... As one of the more innovative crypto projects, Ethereum has seen a historic
    low of about 42 cents and a high of about $4800 in November 2021... Jan 9, 2022 ... A friend of mine
    recently asked me about finding historical open, high, low, and close (OHLC) data for cryptocurrency
    pairs. Jul 27, 2023 ... Source: https://coinmarketcap.com/currencies/ethereum/historical-data/ ...
    Over the past two years (July 9, 2021 to July 9, 2023) ... 2022, when ETH... ETH Historical Price ;
```

$3,747.13 minus $3,875.64. $3,664.40 minus $3,959.28. $4,878.26 22.8%. Nov 10, 2021 (over 2 years).
        $0.433 869871.6%. Oct 20... Historical annualised electricity consumption ... 2021 2022 2023 2024 0
        40 80 120 160 200 240 280 ... The electricity cost parameter plays a crucial role in... 2021 to 2023.
        Bitcoin took less than a month in 2021 to smash its 2020 price record, surpassing $40,000 by Jan. 7,
        2021. By mid-April, Bitcoin prices reached... May 29, 2023, 20.80, 20.97, 20.30, 20.59, 20.59,
        228,295,637. Related Tickers. BTC-USD Bitcoin USD. 68,466.40. -1.09%. ETH-USD Ethereum USD. 3,843.49.
        -0.85%.]The search results indicate that Ethereum reached its all-time high in November 2021, with
        prices briefly surpassing $4800. For a thorough analysis, we need to look at historical data points
        from 2021, 2022, and 2023 to understand the frequency and conditions under which Ethereum has closed
        above $3700.

### Historical Price Data Analysis

#### 1. Reference Classes and Base Rates (Continued)
Let's consider the following data points:
- **All-Time High (ATH)**: Ethereum reached an ATH of approximately $4800 in November 2021.
- **2021**: Ethereum frequently traded above $3700 during the latter part of the year, especially around
    its ATH.
- **2022**: The cryptocurrency market experienced significant volatility, and Ethereum's price dropped
    below $3700 for a considerable period.
- **2023**: The price of Ethereum has been fluctuating, with occasional spikes but generally trading below
    $3700.

Based on this historical data, we can establish a base rate for Ethereum closing above $3700. However, we
    need to adjust for recent trends and market conditions.

#### 2. Recent News and Distinctive Features
- **Technological Developments**: Ethereum's transition to Ethereum 2.0 and the implementation of Proof of
    Stake (PoS) could influence its price positively.
- **Regulatory Environment**: Increased regulation of cryptocurrencies could introduce uncertainty and
    affect market sentiment.
- **Macroeconomic Conditions**: Broader economic factors, such as inflation and interest rates, can impact
    investment in cryptocurrencies.

#### 3. Statistical Modeling
To model Ethereum's potential future price movements, we need to consider its historical volatility and
    distribution patterns. Cryptocurrencies often exhibit high volatility, and their price movements can
    be modeled using distributions like log-normal or power law.

### Combining Estimates
Let's combine our estimates from different approaches:

1. **Base Rate Estimate**:
   - During the peak of the 2021 bull market, Ethereum frequently traded above $3700. Assuming a 30% chance
        based on historical data.

2. **Recent Trends and News**:
   - Positive technological developments and potential regulatory impacts. Adjusting the base rate slightly
        upwards to 35%.

3. **Statistical Modeling**:
   - Considering the high volatility and potential for significant price movements, we can use a log-normal
        distribution to model future prices. However, without precise modeling tools, we estimate a 40%
        chance based on volatility patterns.

### Final Probability Estimate
Let's combine these estimates using a weighted average approach:
- Base Rate: 30%
- Recent Trends and News: 35%
- Statistical Modeling: 40%

Weighted Average Calculation:
$$
\text{Final Probability} = \frac{30\% + 35\% + 40\%}{3} = 35\%
$$

Thought: Do I need to use a tool? No
Final Answer: 0.35The search results indicate that Ethereum reached its all-time high in November 2021,
    with prices briefly surpassing $4800. For a thorough analysis, we need to look at historical data
    points from 2021, 2022, and 2023 to understand the frequency and conditions under which Ethereum has
    closed above $3700.

### Historical Price Data Analysis

#### 1. Reference Classes and Base Rates (Continued)
Let's consider the following data points:
- **All-Time High (ATH)**: Ethereum reached an ATH of approximately $4800 in November 2021.
- **2021**: Ethereum frequently traded above $3700 during the latter part of the year, especially around
    its ATH.
- **2022**: The cryptocurrency market experienced significant volatility, and Ethereum's price dropped
    below $3700 for a considerable period.
- **2023**: The price of Ethereum has been fluctuating, with occasional spikes but generally trading below
    $3700.

Based on this historical data, we can establish a base rate for Ethereum closing above $3700. However, we
    need to adjust for recent trends and market conditions.

#### 2. Recent News and Distinctive Features
- **Technological Developments**: Ethereum's transition to Ethereum 2.0 and the implementation of Proof of
    Stake (PoS) could influence its price positively.

```
- **Regulatory Environment**: Increased regulation of cryptocurrencies could introduce uncertainty and
    affect market sentiment.
- **Macroeconomic Conditions**: Broader economic factors, such as inflation and interest rates, can impact
    investment in cryptocurrencies.

#### 3. Statistical Modeling
To model Ethereum's potential future price movements, we need to consider its historical volatility and
    distribution patterns. Cryptocurrencies often exhibit high volatility, and their price movements can
    be modeled using distributions like log-normal or power law.

### Combining Estimates
Let's combine our estimates from different approaches:

1. **Base Rate Estimate**:
    - During the peak of the 2021 bull market, Ethereum frequently traded above $3700. Assuming a 30% chance
        based on historical data.

2. **Recent Trends and News**:
    - Positive technological developments and potential regulatory impacts. Adjusting the base rate slightly
        upwards to 35%.

3. **Statistical Modeling**:
    - Considering the high volatility and potential for significant price movements, we can use a log-normal
        distribution to model future prices. However, without precise modeling tools, we estimate a 40%
        chance based on volatility patterns.

### Final Probability Estimate
Let's combine these estimates using a weighted average approach:
- Base Rate: 30%
- Recent Trends and News: 35%
- Statistical Modeling: 40%

Weighted Average Calculation:
\[
\text{Final Probability} = \frac{30\% + 35\% + 40\%}{3} = 35\%
\]

Thought: Do I need to use a tool? No
Final Answer: 0.35]

[Finished chain.]
prompt: 2024-04-30: Will ETH close above 3700?
 response: 0.35, index: 6
```

## D.3 GPT-4o Output

```
To forecast whether ETH (Ethereum) will close above $3700 by April 30, 2024, we need to consider several
    factors, including historical price data, market trends, and current events that could influence the
    price. Let's break down the analysis step-by-step.

### Step 1: Historical Data and Base Rates

First, we need to look at the historical price data for ETH to understand its past performance and
    volatility. We can use this data to establish a base rate for how often ETH has closed above $3700 in
    the past.

#### Historical Price Data
 - ETH has been highly volatile, with significant price swings over the past few years.
 - ETH reached an all-time high of around $4800 in November 2021.
 - Since then, the price has fluctuated, with periods of both significant gains and losses. Let's assume we
        have the following historical data for ETH closing prices over the past few years: - 2021: ETH
        closed above $3700 on 50 days.
 - 2022: ETH closed above $3700 on 20 days.
 - 2023 (up to October): ETH closed above $3700 on 10 days. Given this data, we can calculate the base rate
        for ETH closing above $3700.

#### Base Rate Calculation Total number of days in the dataset: 365 (2021) + 365 (2022) + 273 (2023 up to
    October) = 1003 days Total number of days ETH closed above $3700: 50 (2021) + 20 (2022) + 10 (2023) =
    80 days Base rate = 80 / 1003 approximate to 0.08 (8%)

### Step 2: Current Market Trends and News

Next, we need to consider the current market trends and any recent news that could impact the price of ETH.

#### Market Trends

 - The cryptocurrency market has been experiencing increased institutional interest and adoption.
 - Regulatory developments could either positively or negatively impact the market.
 - Macroeconomic factors, such as inflation and interest rates, could influence investor sentiment. ####
        Recent News - Ethereum's transition to a proof-of-stake (PoS) consensus mechanism (Ethereum 2.0)
        could have a significant impact on its price.
 - Any major security breaches or technological advancements could also affect the price. ### Step 3:
        Adjusting the Base Rate Based on the current market trends and recent news, we need to adjust our
        base rate. #### Positive Factors
 - Increased institutional interest and adoption could drive the price higher.
 - The successful implementation of Ethereum 2.0 could boost investor confidence.
```

```
#### Negative Factors
 - Regulatory crackdowns could negatively impact the price.
 - Macroeconomic factors such as rising interest rates could lead to reduced investment in riskier assets
     like cryptocurrencies.

### Step 4: Combining Estimates

We can combine our base rate with adjustments for current trends and news to arrive at a final probability
     estimate.

#### Adjusted Probability

 - Base rate: 8%
 - Positive adjustment: +5% (for increased adoption and Ethereum 2.0)
 - Negative adjustment: -3% (for potential regulatory and macroeconomic risks)
 Final adjusted probability = 8% + 5% - 3% = 10%

### Step 5: Final Probability Estimate

After considering all factors, my best prediction for the probability of ETH closing above $3700 by April
     30, 2024, is: **0.10 (10%)**
This estimate balances the historical base rate with current market trends and potential risks, aiming to
     minimize the Brier score by being well-calibrated and resolved.
```

