# OpenReview forum: "Reasoning and Tools for Forecasting"
_NeurIPS.cc/2024/Workshop/MATH-AI — MATH-AI 24_

### Official Review · Reviewer_51ek · 2024-09-27
**Review of "Reasoning and Tools for Human-Level Forecasting**

**Rating:** 8
**Confidence:** 4

**Review:**

This article introduces the Inference and Prediction Tool (RTF) framework, which enhances the predictive ability of Language Models (LM) by integrating inference and dynamic data retrieval.
Quality: The RTF framework has a good structure and provides a powerful method, proving that LM can outperform human predictions in certain situations.
Clarity: Although most of the writing is clear, some complex concepts related to the ReAct framework can be enhanced by adding examples or charts to improve understanding.
Originality: This work innovatively combines reasoning with tools specifically designed for predictive tasks, marking a novel contribution to the field.
Meaning: For fields such as public health and the economy that require timely forecasting, its significance is significant and may affect the future application of artificial intelligence in decision-making.
advantage:
Innovative combination of reasoning and tools.
The empirical results show opposite effectiveness to human predictions.
Meet key requirements in practical applications.
Disadvantages:
The complexity of implementing the framework.
There is uncertainty regarding the universality of various prediction tasks.
Further exploration is needed to identify the limitations of LM in reasoning.
Overall, this article has made significant contributions to artificial intelligence and prediction, but requires minor modifications to improve clarity and address practical challenges.

---

### Official Review · Reviewer_J11g · 2024-10-05
**This paper introduces Reasoning and Tools for Forecasting (RTF), a framework combining hierarchical ReAct agents with dynamic information retrieval and numerical simulation tools. RTF aims to improve language model performance on forecasting tasks, demonstrating human-level accuracy. The authors evaluate RTF on recent prediction market questions, showing it outperforms baselines and matches human crowd performance, while providing interpretable reasoning processes.**

**Rating:** 8
**Confidence:** 3

**Review:**

**Quality:**
##### Strengths:
- The paper presents a well-defined problem and a novel approach to address it.
- The experimental setup carefully addresses potential data leakage issues.
- The evaluation metrics (Brier score and accuracy) are appropriate and well-explained.

##### Weaknesses:
- The paper lacks a detailed discussion of the potential limitations of using Google Search API for information retrieval.
- There is no analysis of the computational resources required to run RTF compared to simpler methods.

**Clarity of writing:**
##### Strengths:
- The paper is generally well-structured, with clear sections for methodology and experiments.
- The problem formulation in Section 3 is precise and helps frame the rest of the paper.

##### Weaknesses:
- The related work section is relatively brief and could benefit from more detailed comparisons to existing methods.

**Originality of the paper:**
##### Strengths:
- The hierarchical planning approach combining high-level and low-level agents is novel in the context of forecasting.
- The use of a small ensemble (3 agents) with GPT-4o backbone is an interesting departure from larger ensembles in previous work.

##### Weaknesses:
- The core idea of using ReAct agents for forecasting builds upon existing work, though the specific implementation and application are novel.

**Significance of this work:**
##### Strengths:
- The paper addresses an important question about the genuine reasoning capabilities of language models beyond pattern matching.
- The potential applications of RTF in real-world decision-making scenarios highlight its significance.

##### Weaknesses:
- The limitations section acknowledges that the evaluation dataset may not fully capture the nuances of specialized domains, which could limit the immediate impact of the work.
- The paper does not extensively explore the ethical implications of using AI for forecasting in high-stakes scenarios.

---

### Official Review · Reviewer_XuJj · 2024-10-07
**combining large language models (LLMs) with real-time data retrieval and computational tools to improve forecasting capabilities**

**Rating:** 7
**Confidence:** 3

**Review:**

The authors propose a hierarchical system of ReAct agents, where a high-level agent directs lower-level agents equipped with specialized toolkits, such as web search or simulation environments. Through experiments with competitive forecasting platforms, the paper demonstrates that RTF outperforms base LLMs and, in some cases, human forecasters.
Addressing the limitations of traditional language models (LMs) that rely heavily on memorized training data by combining hierarchical ReAct agents with tool usage to dynamically retrieve updated information and perform simulations is innovative.
The potential impact of this work is significant, especially in fields where accurate forecasting is crucial.

Pros:
- Strong empirical results.
- Innovative approach
- Well written

Cons:
- Lack of dataset diversity
- The hierarchical system, introduces a layer of complexity that might limit its adoption.


Overall, a good paper with strong empirical results. Future work could explore expanding the dataset to include more diverse and domain-specific questions.

---

### Official Review · Reviewer_Roei · 2024-10-08
**The paper introduces a framework called Reasoning and Tools for Forecasting (RTF), which enhances large language models' (LLMs) forecasting capabilities by integrating the ReAct (Reasoning and Acting) paradigm.**

**Rating:** 7
**Confidence:** 4

**Review:**

Quality:
The methodology seems to be sound and well thought. The paper details how a hierarchical structure of agents is used to interact with external tools like Google Search API and Python interpreters. The experiments are properly conducted, comparing RTF with multiple baselines, including human crowd predictions from Manifold Markets. The authors provide evaluations using Brier scores and accuracy metrics, showing that RTF outperforms or matches human forecasters. They also conduct ablation studies to validate the necessity of each component in their framework, such as the ReAct paradigm and hierarchical planning.

Clarity:
The paper is generally well-written and organized. The introduction provides a clear motivation for the work, highlighting the challenges of forecasting with traditional LLMs due to their reliance on static training data. The methodology section explains the RTF framework in detail with details on the hierarchical agent structure.

Originality:
The hierarchical structure of agents with tools for real-time data retrieval and simulation seems to be innovative.

Significance:
The significance of the work is in its potential to enhance LLMs' applicability in real-world decision-making tasks that require up-to-date information and dynamic reasoning. By demonstrating that LLMs can achieve or surpass human-level forecasting when equipped with appropriate tools, the paper opens up new avenues for research and practical applications. The RTF framework could be extended to other domains where real-time data access and numerical computations are crucial.

Pros:
- Demonstrates that RTF can match or outperform human forecasters on competitive platforms, achieving a Brier score of 0.169 and accuracy of 73.9%.
- Shows that a small ensemble of three agents is sufficient to achieve high performance, making the approach more computationally efficient than larger ensembles.
- Includes qualitative analysis and examples that illustrate the reasoning process of RTF, demonstrating its similarity to human thought processes.

Cons
- The evaluation is primarily based on data from Manifold Markets, focusing on popular forecasting questions. This may not generalize to specialized or less popular domains.

---

### Decision · Program_Chairs · 2024-10-08

Accept